

# Effects of root exudates of woody species on the soil anti-erodibility in the rhizosphere in a karst region, China

Zhen Hong Wang[1], Hong Fang[2] and Mouhui Chen[3]

[1] School of Environmental Science and Engineering, Chang'an University, Xi'an, China
[2] Water-affair Authority, Xifeng County, Guiyang, Guizhou, China
[3] College of Forestry, Guizhou University, Guiyang, Guizhou, China

## ABSTRACT

**Introduction.** Rhizospheres, the most active interfaces between plants and soils, play a central role in the long-term maintenance of the biosphere. The anti-erodibility of soils (AES) regulated by the root exudates is crucial to the soil stability in the rhizospheres. However, scientists still debate (1) the key organic matter of the root exudates affecting the AES and (2) the interspecific variation of these root exudates.

**Methods.** We used an incubation of soils to test the effects of the root exudates from eight woody plant species on the change in soil aggregation and identified the organic matter in these root exudates with gas chromatography-mass spectrometry (GC-MS) and biochemical methods. Furthermore, the relationships between the organic matter in the exudates and the AES in the rhizospheres of 34 additional tree species were analyzed.

**Results.** The water-stable aggregates of the soils incubated with the root exudates increased by 15%–50% on average compared with control samples. The interspecific differences were significant. The root exudates included hundreds of specific organic matter types; hydrocarbon, total sugar, total amino acids, and phenolic compounds were crucial to the AES. These organic matter types could explain approximately 20–75% of the variation in the total effect of the root exudates on the AES, which was quantified based on the aggregate status, degree of aggregation, dispersion ratio, and dispersion coefficient.

**Discussion.** The effects of the root exudates on the AES and the interspecific variation are as important as that of root density, litters, and vegetation covers. Many studies explored the effects of root density, litters, vegetation covers, and vegetation types on the AES, but little attention has been paid to the effects of the root exudates on the AES. Different plants secrete different relative contents of organic matter resulting in the variation of the effect of the root exudates on the AES. Our study quantified the causal relationships between the root exudates and the AES using modeling experiments in laboratory and field observations and indicated the interspecific variation of the AES and organic matter of the root exudates.

**Conclusions.** More organic compounds of the exudates related to the AES were recognized in this study. These results enhance the understanding of the soil stability at a slope and can be applied to ecosystem restoration.

Corresponding authors
Zhen Hong Wang,
w_zhenhong@126.com
Hong Fang,
fanghong20072008@126.com

## INTRODUCTION

The rhizosphere, a term firstly used by Hiltner, is a zone of soil surrounding the root. The root is directly affected by the rhizosphere (*Gregory, 2006*). The size of the rhizosphere differs spatially and temporally depending on the factors considered. It ranges from a fraction of a millimeter for microbial populations and immobile nutrients to tens of millimeters for mobile nutrients and exudates released from roots (*Gregory, 2006*). The rhizosphere differs from the bulk soil due to a range of biological, chemical, and physical processes that occur as a consequence of root growth, release of exudates, and rhizodeposition (*Kandeler et al., 2002*; *Marschner & Baumann, 2003*; *Hinsinger et al., 2005*). When the seeds germinate and roots grow through the soil, the release of the root exudates changes all these processes; the soil particles adhere to each other to form soil aggregates, the nutrient availability of organic anions increases, and signaling molecules selectively induce the multiplication of microbes (*Whipps, 2001*). The root exudates therefore directly provide the driving forces for the development of the soil structure (*Walker et al., 2003*). The rhizosphere formation and corresponding soil stability in the rhizosphere greatly contribute to the stability of the slope, carbon sequestration, and maintenance of the biosphere; thus, the rhizosphere is partially responsible for the unique characteristics of Earth as compared to other planets (*Walker et al., 2003*; *Kuzyakov, Hill & Jones, 2007*; *Vannoppen et al., 2015*).

Root exudates affect the anti-erodibility of soils (AES) as significantly as the root density, litters, and vegetation cover (*Fattet et al., 2011*; *Vannoppen et al., 2015*). These exudates play several roles in directly and indirectly strengthening the AES: (1) The adhesive properties of the root exudates bind the soil particles together to enhance the formation of water-stable aggregates (*Bronick & Lal, 2005*; *De Baets et al., 2008*); (2) The release of root exudates is a continual source of organic matter, which will improve the soil structure with respect to the size, shape, and arrangement of solids and voids, continuity of pores and voids, and their capacity to retain and transmit fluids and organic and inorganic substances (*Lal, 1991*); (3) The aggregate formation and stability are indirectly influenced by microorganisms, which feed on the root exudates and produce hypha and polysaccharides to bind soil particles together (*Andrade et al., 1998*). The strengthened AES increases the resistance to erosion by raindrops, surface runoff, concentrated flow, and seepage flow at the root–soil interface (*Vannoppen et al., 2015*).

Many studies have dealt with these roles. For example, *Tisdall & Oades (1982)* found that the water-stable aggregates (>0.25 mm) depend on the root exudates and fungal hyphae. The stability of microaggregates was determined using the content of persistent organo-mineral complexes and transient polysaccharides. *Czarnes, Dexter & Bartoli (2000)* mixed bacterial xanthan and an analogue of root mucilage (polygalacturonic acid) with soils to simulate the adhesive effects of the root exudates and suggested that xanthan

and polygalacturonic acid increase the tensile strength of the soils. Subsequently, it has been reported that rhizosphere soils contain larger pores than bulk soils (*Whalley et al., 2005*). The soils that adhered to maize roots in silty soil have greater aggregation strength (450–500 kPa) than the soils that did not (410–420 kPa; *Czarnes, Dexter & Bartoli, 2000*). Many studies also indicated that mycorrhizal hyphae were involved in the adhesion of soil particles to roots, in combination with root hairs, immature xylem vessels, and the mucilage from roots, resulting in the formation of rhizosheaths (*Amellal et al., 1998*; *McCully, 1999*; *Young & Crawford, 2004*). In addition, microbial biomass carbon, hot-water soluble carbohydrate carbon, and soil organic carbon are assumed to constitute the root exudates, leading to the formation of soil aggregates. The experimental results indicated that the chemical bonding of these compositions accounts for 14.7% of the variation in the macroaggregates (>0.212 mm), while the physical binding of the root systems accounts for 39.0% (*Jastrow, Milier & Lussenhop, 1998*; *Wang et al., 2014a*; *Wang et al., 2014b*).

Overall, these studies advanced the field primarily based on the following aspects: (1) The root exudates were stimulated with analogues. The soil sample mixtures and stimulated root exudates were incubated to test the effects of the root exudates on the AES (*Morel et al., 1991*); (2) The root exudates collected from one or two annual crop plants were mixed with the soil samples. The mixtures were incubated to identify the effects of the root exudates on the AES; (3) The effect of the root exudates on the AES was theoretically or experimentally separated from that of other factors such as the root density, litters, and vegetation cover. However, it still remains unclear what types of organic matter of the root exudates in the rhizosphere of different woody plants are crucial to the AES. The interspecific variation of the organic matter and the AES in the rhizosphere of these woody plants are also unknown. Moreover, while the effect of the root exudates has been identified in the laboratory, some caution is required in extrapolating the results to field conditions (*Gregory, 2006*). Additionally, karst regions in which concealed erosion is a primary way of soil erosion are characterized by strong karstification (*Wang et al., 2014*; *Wang et al., 2014b*). Understanding the relationship between the AES and root exudates is especially important to control the concealed erosion, which is the vertical movement of soil particles due to a seepage flow, with little surface runoff. However, little attention has been paid to this relationship.

We conducted incubation experiments and extrapolation experiments in this study to clarify the above-mentioned aspects (Fig. 1). We primarily focus on the following questions: (1) How do the root exudates affect the water-stable aggregates, microaggregates, mean weight diameter (MWD), and geometric mean diameter (GMD) of the soil subsamples when these are respectively incubated with the root exudates from eight tree species? Which of the organic matter compounds in the root exudates are more closely related to the MWD and GMD; and (2) How do the test results for the organic matter in the root exudates and the comprehensive indices of the AES (aggregation status, degree of aggregation, dispersion ratio, and dispersion coefficient) in the rhizosphere soils change when the number of tested

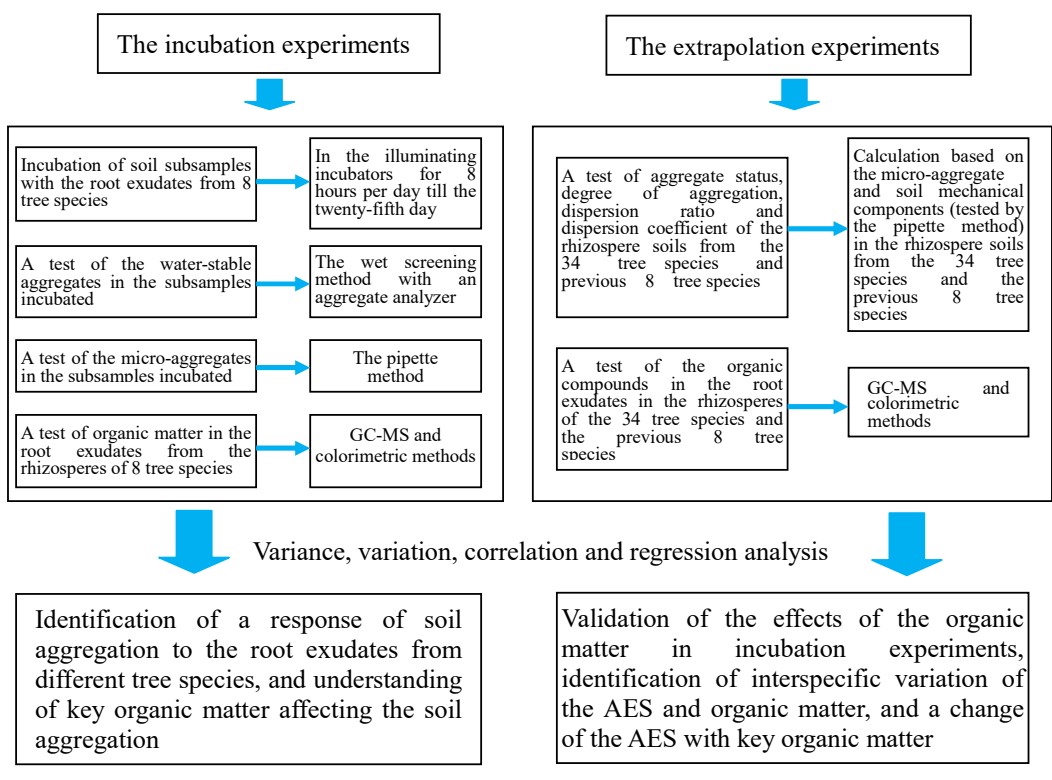

**Figure 1   Flowchart of the study.**

tree species is increased to 34 species (in addition to the previous eight plants)? How does the key organic matter affect the comprehensive indices of the AES?

## MATERIALS AND METHODS

### Root exudate extraction

We selected eight of the typical tree species to extract the root exudates in a karst forest of the Qianling Mountains (106°41′–106°42′E, 26°17′–26°22′N, 1,100–1,396 m elevation) in Guiyang, China. These species included *Carpinus pubescens, Cladrastis platycarpa, Zanthoxylum planispinum Sieb.et Zucc., Ligustrum lucidum, Itea yunnanensis, Cinnamomum glanduliferum, Cyclobalanopsis gracilis,* and *Platycarya longipes.* We further determined three sample trees for each species based on the similarity in the individual growth and the lack of diseases, pests, and anthropogenic disturbance impacting the individual growth. The litters, humus layers, and soils under the canopy of each sample tree were then removed. After finding the living fibrous roots, we removed the external (thickness >1 cm) soils around the roots and collected the inner soils (0–1 cm thickness; at least 500 g). The subsamples, which equaled 60 g of dried soils, were taken from the 500-g soil samples and placed in a wide mouth bottle. The root exudates of the subsamples were extracted with 200 ml ether under oscillating conditions at 20 °C (for 1 h). The mixture in the wide mouth bottle was then filtered and the filtrate was condensed at 20 °C using a rotary evaporator. Finally, the condensed filtrate was diluted to 10 ml to obtain the mother
liquid of the root exudates. The remaining soil samples were sieved through 2-mm and 0.25-mm sieves after open-air drying for a week. The soil samples were then used to test the organic matter of the root exudates. All field experiments of this study have been approved by the Administration Bureau of Two Lakes and One Reservoir in Guiyang.

## Soil incubation and AES tests

Approximately 2 kg of samples of rendzina soil were collected from a depth of 0–20 cm in an evergreen broadleaf forest (elevation: 1,220 m) in Guiyang. After open-air drying, the soil samples were sieved through a 2-mm sieve. We then weighed three soils samples of 30 g, which were put into three 100-ml conical flasks. We successively added one, two, and three times the volume (10 ml) of the mother liquid from one tree species to the conical flasks. The different volumes of the mother liquid were replicated three times. The control samples were prepared by adding the same volume of distilled water to three 100-ml conical flasks with 30 g of soil sample. We then adjusted the C/N ratio of the mixtures in all conical flasks to 10 using $KNO_3$ solution (the amount of $KNO_3$ solution was calculated based on the C and N contents of the soil samples and the mother liquid of the root exudates, which were determined in advance using the potassium dichromate oxidation method and Kjeldahl determination, respectively). The water content of the mixtures was also adjusted to ~60% of the field moisture capacity. Subsequently, all conical flasks were closed with rubber stoppers and were incubated at 25 °C in illuminating incubators for 8 h every day for 25 days. The water-stable aggregates and microaggregates of the soils were then tested. The water-stable aggregates were analyzed using the wet-screening method and an aggregate analyzer, which included the five particle diameters: >2mm, 2–1 mm, 1–0.5 mm, 0.5–0.25 mm and <0.25 mm. The microaggregates were measured with the pipette method (0.25–0.05 mm, 0.05–0.02 mm, 0.02–0.002 mm and <0.002 mm). The MWD and GMD were calculated using Eqs. (1) and (2).

$$MWD = \frac{\left(\sum_{i=1}^{n} \bar{X}_i W_i\right)}{\sum_{i=1}^{n} W_i} \tag{1}$$

$$GMD = \exp\left(\frac{\sum_{i=1}^{n} W_i \ln\bar{X}_i}{\sum_{i=1}^{n} W_i}\right) \tag{2}$$

where $\bar{x}_i$ is the mean diameter (mm) of the aggregates within a particle-size range and $W_i$ is the ratio of the weight (g) of the aggregates within the particle-size range to the total weight (g) of the soil sample.

## GC-MS analysis of the organic matter and active biological matter tests

We sieved 40 g of the soil samples from the rhizospheres of each tree species through a 40-mesh sieve and filled the soil in a 500-ml conical flask with a stopper. Subsequently, 150 ml of dichloromethane was decanted into the conical flask. The conical flask was closed with a stopper and continually oscillated for 1 h. Subsequently, the mixture in the conical flask was extracted for 20 min using ultrasonic waves and filtrated. The residue was collected and placed into another 500-ml conical flask. The root exudates of the residue were extracted in the same way. The filtrates from two extractions were mixed, concentrated for 20 min with

a rotary evaporator, and dissolved with 5 ml ether that was led through a 0.45-$\mu$m filter membrane. Finally, the mixed liquid was filled into a sterile centrifuge tube for GC-MS analysis.

The GC-MS analysis of the organic matter was performed using a HP 6890 gas chromatograph equipped with an Agilent MSD 5975C mass spectrometer (Agilent Technologies) and chromatographic column (AB-5MS 5% phenyl-95% dimethylpolysiloxane; 30 m $\times$ 0.25 mm $\times$ 0.25 $\mu$m; elastic quartz capillary). The temperature in the vaporization chamber was maintained at 250 °C. Highly pure helium was used as the carrier gas, with a flow rate of 1.0 ml min$^{-1}$. The inlet pressure was 7.62 psi; the split ratio was 20:1. The solvent delay time was set to 1.5 min (*Müller et al., 2008*; *Hutzler et al., 2014*). The identification of the organic matter and relative contents was conducted with a mass spectrometry data system. Specifically, the different peaks of the total ion spectrum were first compared with the standard spectrum of the NIST05 and Wiley275 databases to determine the volatile constituents of the root exudates. The peak area normalization method was then used to measure the relative mass fraction of the volatile constituents.

We tested the active biological matter of the rhizosphere soils, including total sugar, total amino acids, phenolic compounds, and free amino acid, using anthracenone colorimetry (*Abdelhamid et al., 2013*), tri-ketone colorimetry (*Song et al., 2009a*; *Song et al., 2009b*), and Folin-ciocalteu colorimetry (*Song et al., 2009a*; *Song et al., 2009b*; *Faujdar, Prasad & Paliwal, 2012*).

## Tests and analysis of additional plant species

We selected 34 additional tree species (different from the eight tree species) of the karst forests of Guiyang to validate the effect of the root exudates. Specifically, we used the same methods as described above to collect the rhizosphere soils of three sample trees for each species, extract the root exudates, and identify the contents of the organic matter with GC-MS (only 13 tree species showed satisfactory flow diagrams). The active biological matter was detected with biochemical methods. The four comprehensive AES indices, aggregate status, degree of aggregation, dispersion ratio, and dispersion coefficient, of the rhizosphere soils of the 34 species and the previous 8 plants were quantified using the Eqs. (3)–(6) (*Wang et al., 2014a*; *Wang et al., 2014b*). Finally, the relationship between the AES indices and the organic matter in the root exudates was analyzed.

Aggregate status (%) = (the micro-aggregate at a $>$ 50 $\mu$m particle diameter, %)

$-$ (soil mechanical components at a $>$ 50 $\mu$m particle diameter, %)       (3)

$$\text{Degree of aggregation (\%)} = \frac{\text{Aggregate status} \times 100}{\text{The micro-aggregates at a} > 50 \ \mu\text{m particel diameter}} \quad (4)$$

$$\text{Dispersion ratio (\%)} = \frac{\text{The micro-aggreate at a} < 50 \ \mu\text{m particle diameter} \times 100}{\text{Soil mechanical components at a} < 50 \ \mu\text{m particle diameter}} \quad (5)$$

Dispersion coefficient (%)

$$= \frac{\text{The micro-aggreate at a} < 2 \ \mu\text{m particle diameter} \times 100}{\text{Soil mechanical components at a} < 2 \ \mu\text{m particle diameter}} \quad (6)$$

## Data analysis

The $t$-test was used to measure the differences between the control samples (i.e., test of single population) and the water-stable aggregates, microaggregates, MWD, and GMD, respectively. Specifically, we used the formula $t = (\bar{\mu} - \mu_0)\sqrt{n-1}/s$ where $\bar{\mu}$ and $\mu_0$ were the mean indices of the AES incubated with the root exudates and the distilled water (controls), respectively; $s$ is the standard deviation; and $n$ is the number of samples. A variation coefficient ($CV$) was used to test the interspecific variation of the individual indices of the AES ($F$-and $T$-tests could not be applied); $CV(\%) = s \times 100/\bar{\mu}$. When $CV \geqslant 30\%$, it was statistically defined that there was a significant statistical difference among plant species; when $CV < 30\%$, the significance level was determined by $CV_u$. The $CV_u$ is an upper confidence limit of the $CV$. When $CV < CV_u$, no significant statistical variation between the plant species was observed; Here, the $CV_u = \{(X^2_{1-\alpha}(n-1)[1+CV^2(n-1)/n)]\}/[(n-1)CV^2]$, where $X^2_{1-\alpha}(n-1)$ was obtained by searching the quantiles of the chi-squared distribution when the free degree $= n-1$ and probability $1-\alpha$ (*Standardization Administration of PRC, 2009*).

The interspecific differences of the relative contents of the organic matter identified by GC-MS and the active biological matter were also tested using $CV$. The AES of the rhizosphere and non-rhizosphere soils was compared using the $t$-test of a double-population. The relationship between the four comprehensive AES indices and the organic compounds was described with Pearson's correlation coefficients. The significance levels were also tested with a $t$-test.

# RESULTS

## Effects of the root exudates on the AES

The comparison with the control samples shows that the amount of water-stable aggregates (>2 mm and 2–1 mm) of the soils incubated with 1–3 times of the mother liquid of the root exudates increased, except for a few of the soil samples (Table 1). The average increase is 15.52% and 21.39% (1×), 13.33% and 35.58% (2×), and 19.25% and 40.65% (3×) for the two aggregate diameters, respectively. The rhizosphere soils of *C.platycarpa*, *C.gracilis*,*I.yunnanensis* and *P.longipes* contain more water-stable aggregates (>2 mm and 2–1 mm) than other plants. However, the incubation of the soils with 1–3 times of the mother liquid resulted in the decrease of the concentration of water-stable aggregates (<0.25 mm) of 41.3% (1×), 51.34% (2×) and 58.30% (3×), respectively (Table 1). The water-stable aggregates of the soils incubated with 2–3 times of the mother liquid did not always show a higher percentage. The $t$-test indicated a significant difference between the water-stable aggregates (>2 mm, 2–1 mm, and <0.25 mm) and the control samples. The water-stable aggregates with diameters of 0.5–0.25 mm show a relatively small change compared to the control samples, although the $t$-test implied a significant difference. The water-stable aggregates with diameters of 1–0.5 mm did not indicate a notable difference between the incubated soils and control samples (Table 1). The $CV$ and $CV_u$ revealed that the water-stable aggregates of all diameters significantly differ among the tree species; the aggregates with a diameter of 1–0.5 mm or 0.5–0.25 mm show a greater difference than the aggregates with other diameters (Table 1).

**Table 1** Compositions of the water-stable aggregates of the soils samples incubated with the root exudates from the eight plant species.

| Tree species | Concentrations | The water-stable aggregates at different aggregate diameters (%) | | | | |
|---|---|---|---|---|---|---|
| | | >2 mm | 2-1 mm | 1–0.5 mm | 0.5–0.25 mm | <0.25 mm |
| Controls | 0 | 14.26 ± 1.07 | 16.95 ± 1.49 | 28.13 ± 1.07 | 7.10 ± 0.28 | 33.56 ± 0.86 |
| Carpinus pubescens | 1×* | 18.18 ± 0.20 | 17.30 ± 0.30 | 25.87 ± 1.70 | 8.10 ± 0.78 | 30.56 ± 1.54 |
| | 2× | 14.94 ± 0.46 | 22.23 ± 0.91 | 29.09 ± 1.74 | 6.99 ± 0.85 | 26.76 ± 1.73 |
| | 3× | 17.41 ± 1.34 | 20.84 ± 0.83 | 27.17 ± 0.80 | 8.16 ± 0.42 | 26.42 ± 1.66 |
| Cladrastis platycarpa | 1× | 14.93 ± 1.87 | 22.57 ± 2.08 | 26.67 ± 0.94 | 8.35 ± 0.50 | 27.48 ± 1.10 |
| | 2× | 20.53 ± 1.11 | 23.91 ± 1.61 | 23.66 ± 3.02 | 7.00 ± 0.92 | 24.91 ± 1.50 |
| | 3× | 18.81 ± 0.49 | 23.07 ± 1.34 | 24.97 ± 1.75 | 7.64 ± 0.44 | 25.50 ± 1.38 |
| Zanthoxylum Planispinum Sieb.et Zucc. | 1× | 14.12 ± 0.66 | 25.33 ± 1.36 | 25.05 ± 1.75 | 8.23 ± 1.37 | 27.28 ± 0.66 |
| | 2× | 15.82 ± 0.64 | 20.32 ± 2.35 | 28.27 ± 2.09 | 6.85 ± 0.79 | 28.76 ± 0.68 |
| | 3× | 15.90 ± 1.74 | 22.50 ± 2.63 | 28.15 ± 0.72 | 7.50 ± 0.15 | 25.95 ± 1.07 |
| Ligustrum lucidum | 1× | 17.89 ± 0.29 | 17.42 ± 2.14 | 28.43 ± 0.75 | 9.19 ± 0.79 | 27.06 ± 2.31 |
| | 2× | 17.17 ± 0.90 | 15.40 ± 1.27 | 29.88 ± 2.29 | 9.11 ± 0.38 | 28.45 ± 1.10 |
| | 3× | 16.42 ± 1.14 | 23.13 ± 1.06 | 26.34 ± 0.91 | 8.11 ± 1.07 | 26.00 ± 1.47 |
| Itea yunnanensis | 1× | 16.68 ± 1.58 | 19.85 ± 1.26 | 27.79 ± 1.33 | 10.38 ± 0.96 | 25.30 ± 1.00 |
| | 2× | 14.05 ± 0.38 | 22.40 ± 0.71 | 27.85 ± 1.48 | 10.08 ± 1.27 | 25.62 ± 1.30 |
| | 3× | 18.28 ± 2.03 | 25.76 ± 1.50 | 24.48 ± 0.82 | 8.70 ± 0.96 | 22.77 ± 2.10 |
| Cinnamomum glanduliferum | 1× | 13.99 ± 1.40 | 20.75 ± 2.40 | 30.24 ± 1.29 | 8.39 ± 0.49 | 26.63 ± 0.92 |
| | 2× | 13.97 ± 0.30 | 24.63 ± 0.61 | 27.79 ± 0.34 | 6.89 ± 0.32 | 26.72 ± 0.84 |
| | 3× | 14.78 ± 0.86 | 23.01 ± 1.13 | 26.81 ± 0.92 | 9.91 ± 0.48 | 25.48 ± 1.59 |
| Cyclobalanopsis gracilis | 1× | 19.11 ± 3.64 | 18.26 ± 1.96 | 27.87 ± 2.35 | 6.57 ± 0.16 | 28.19 ± 1.21 |
| | 2× | 19.41 ± 0.73 | 19.96 ± 1.82 | 28.25 ± 1.55 | 6.64 ± 1.03 | 25.75 ± 1.75 |
| | 3× | 19.95 ± 1.68 | 20.20 ± 1.84 | 28.70 ± 0.62 | 7.05 ± 0.52 | 24.10 ± 0.60 |
| Platycarya longipes | 1× | 16.87 ± 1.06 | 18.52 ± 0.64 | 27.41 ± 0.51 | 8.35 ± 0.81 | 28.85 ± 0.75 |
| | 2× | 13.38 ± 0.90 | 27.34 ± 2.42 | 28.29 ± 1.17 | 8.03 ± 0.81 | 22.95 ± 0.91 |
| | 3× | 14.45 ± 1.91 | 23.45 ± 2.53 | 29.21 ± 1.51 | 7.13 ± 0.70 | 25.76 ± 0.88 |
| T-test | 1× | T = 3.03, p < 0.01 | T = 2.89, p < 0.025 | T = 1.19, p > 0.05 | T = 3.33, p < 0.01 | T = 9.93, p < 0.005 |
| | 2× | T = 1.90, p < 0.05 | T = 3.75, p < 0.005 | T = 0.36, p > 0.05 | T = 1.26, p > 0.05 | T = 10.26, p < 0.005 |
| | 3× | T = 3.71, p < 0.005 | T = 9.06, p < 0.005 | T = 1.81, p > 0.05 | T = 2.61, p < 0.025 | T = 18.18, p < 0.005 |

Wang et al. (2017), *PeerJ*, DOI 10.7717/peerj.3029

**Table 1** (*continued*)

| Tree species | Concentrations | The water-stable aggregates at different aggregate diameters (%) | | | | |
|---|---|---|---|---|---|---|
| | | >2 mm | 2-1 mm | 1–0.5 mm | 0.5–0.25 mm | <0.25 mm |
| Coefficientsof variaiton (*CV*,%) | *CV* of 1× | 11.74 | 13.99 | 5.83 | 12.66 | 5.68 |
| | $CV_u$ | 1.29 | 1.28 | 1.30 | 1.29 | 1.30 |
| | *CV* of 2× | 16.42 | 16.27 | 6.60 | 16.47 | 7.20 |
| | $CV_u$ | 1.28 | 1.28 | 1.30 | 1.28 | 1.29 |
| | **CV** of 3× | 11.50 | 7.45 | 6.28 | 11.74 | 4.80 |
| | $CV_u$ | 1.29 | 1.29 | 1.30 | 1.29 | 1.31 |

**Notes.**

*1×, 2×and 3×represent the treatments incubated by 1, 2, and 3 times of the mother liquid of the root exudates. The same below.

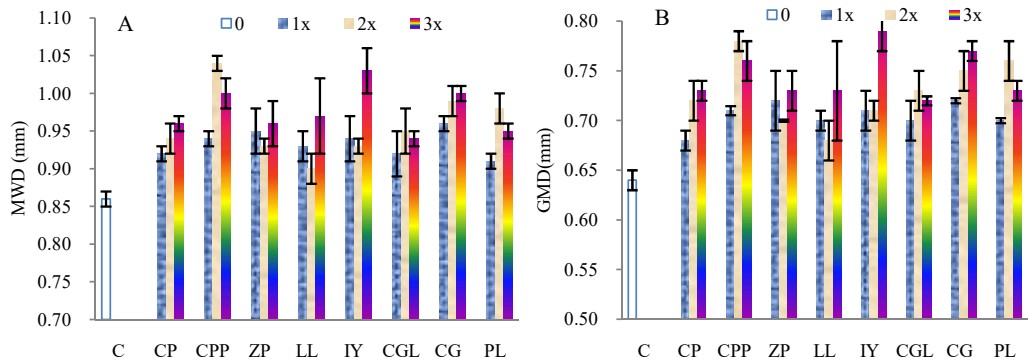

**Figure 2** **Mean weight diameter (MWD, A) and geometric mean diameter (GMD, B) of the soils incubated with the root exudates from the eight tree species.** C, Control sample; CP, *C.pubescens*; CPP, *C.platycarpa*; ZP, *Z. planispinum Sieb.et Zucc.*; LL, *L.lucidum*; IY, *I.yunnanensis*; CGL, *C.glanduliferum*; CG, *C.gracilis*; PL, *P. longipes*.

The MWD of the soils incubated with 1–3 times of the mother liquid increased on average by 8.58%, 11.34%, and 13.52%, respectively, compared with the control samples (Fig. 2A). The $t$-test indicated significant differences ($1\times$ : $t = 11.58$, $p < 0.005$, $n = 8$; $2\times$: $t = 5.86$, $p < 0.005$, $n = 8$; $3\times$: $t = 10.03$, $p < 0.005$, $n = 8$). The MWDs of the soils incubated with the root exudates extracted from the rhizosphere soils of *C.platycarpa*, *I. yunnanensis* and *C. gracilis* are relatively greater than that of the other tree species. The GMD of the soils increased by 10.16%, 13.87%, and 16.41%, respectively, compared with the control samples (Fig. 2B).The increasing rates are higher than that of the MWD. The $t$-test indicated significant differences ($1\times$: $t = 13.13$, $p < 0.005$, $n = 8$; $2\times$: $t = 7.09$, $p < 0.005$, $n = 8$; $3\times$: $t = 11.08$, $p < 0.005$, $n = 8$). The GMD of the soils incubated with the root exudates from the rhizospheres of *C.platycarpa*, *I. yunnanensis* and *C. gracilis* are greater than that of the other plant species (Fig. 2B). However, the values of the MWD and GMD did not always increase with increasing volume of the mother liquid.

Compared with the control samples, the microaggregates (0.05–0.02 mm) of the soils incubated with 1–3 times the volume of the mother liquid decreased by 46.42% ($1\times$), 54.72% ($2\times$), and 36.01% ($3\times$), respectively. However, the microaggregates (>0.002 mm) only decreased by 10.70%, 15.34%, and 21.59%, respectively (Table 2). Conversely, the microaggregate concentration (0.02–0.002 mm) strongly increased; the increasing rate is 129.85%, 135.68%, and 157.1%, respectively. The $t$-test showed no significant differences between the microaggregates with a diameter of 2–0.25 mm or 0.25–0.05 mm and the control samples (Table 2). Comparatively, the differences are more significant between the microaggregates with diameters of 0.05–0.02 mm, 0.02–0.002 mm, or <0.002 mm and the control samples. Based on *CV* and *CV*$_u$, the microaggregates with different diameters significantly differ among the tree species, except for the microaggregate (2–0.25 mm) of the soils incubated with two times the volume of the mother liquid. The *CV* of the microaggregates with a diameter of 0.05–0.02 mm or 0.02–0.002 mm is relatively large.

Wang et al. (2017), *PeerJ*, DOI 10.7717/peerj.3029

**Table 2  Microaggregate components of the soil samples incubated with the root exudates from the eight plant species.**

| Tree species | Concentrations | The micro-aggregates at different aggregate diameters(%) | | | | |
|---|---|---|---|---|---|---|
| | | 2–0.25 mm | 0.25–0.05 mm | 0.05–0.02 mm | 0.02–0.002 mm | >0.002 mm |
| Controls | 0 | 71.87 ± 1.94 | 12.45 ± 0.84 | 5.72 ± 1.07 | 2.27 ± 0.52 | 7.69 ± 1.00 |
| *Carpinus pubescens* | 1× | 71.35 ± 2.09 | 12.29 ± 1.57 | 4.92 ± 0.48 | 4.44 ± 1.25 | 7.00 ± 0.60 |
| | 2× | 73.82 ± 1.23 | 12.62 ± 1.37 | 2.52 ± 0.79 | 4.60 ± 0.85 | 6.44 ± 1.08 |
| | 3× | 70.11 ± 1.65 | 13.69 ± 2.41 | 3.72 ± 0.44 | 6.08 ± 0.27 | 6.40 ± 1.03 |
| *Cladrastis platycarpa* | 1× | 72.28 ± 2.17 | 13.92 ± 1.26 | 1.44 ± 0.53 | 5.52 ± 1.34 | 6.84 ± 0.47 |
| | 2× | 71.53 ± 0.88 | 13.11 ± 0.61 | 2.60 ± 0.08 | 6.12 ± 0.92 | 6.64 ± 0.75 |
| | 3× | 70.94 ± 2.22 | 10.98 ± 0.92 | 5.04 ± 1.32 | 7.24 ± 1.65 | 5.80 ± 1.04 |
| *Zanthoxylum Planispinum Sieb.et Zucc.* | 1× | 69.97 ± 2.71 | 13.99 ± 0.65 | 4.08 ± 0.65 | 5.16 ± 1.16 | 6.80 ± 1.43 |
| | 2× | 74.16 ± 2.25 | 12.04 ± 2.24 | 3.52 ± 1.61 | 4.88 ± 1.54 | 5.40 ± 0.74 |
| | 3× | 70.73 ± 2.53 | 12.55 ± 1.70 | 5.00 ± 0.64 | 5.56 ± 0.53 | 6.16 ± 1.03 |
| *Ligustrum lucidum* | 1× | 74.51 ± 4.12 | 11.01 ± 1.95 | 2.16 ± 0.36 | 6.48 ± 0.70 | 5.84 ± 1.32 |
| | 2× | 72.15 ± 2.16 | 13.13 ± 0.47 | 3.12 ± 0.42 | 5.44 ± 0.82 | 6.16 ± 0.82 |
| | 3× | 72.57 ± 0.91 | 11.51 ± 1.20 | 3.60 ± 0.79 | 6.60 ± 0.98 | 5.72 ± 0.96 |
| *Itea yunnanensis* | 1× | 75.35 ± 2.63 | 10.37 ± 0.30 | 3.56 ± 0.76 | 4.22 ± 0.52 | 6.50 ± 1.25 |
| | 2× | 73.78 ± 3.13 | 12.54 ± 2.01 | 3.24 ± 0.26 | 4.36 ± 1.16 | 6.08 ± 1.31 |
| | 3× | 74.09 ± 0.91 | 11.07 ± 1.10 | 2.88 ± 0.73 | 5.76 ± 0.33 | 6.20 ± 1.46 |
| *Cinnamomum glanduliferum* | 1× | 72.60 ± 1.93 | 10.61 ± 0.69 | 3.36 ± 0.30 | 6.64 ± 1.88 | 6.80 ± 0.55 |
| | 2× | 73.29 ± 1.47 | 14.35 ± 1.62 | 1.40 ± 0.23 | 3.68 ± 0.57 | 7.28 ± 0.58 |
| | 3× | 73.10 ± 2.72 | 12.78 ± 2.00 | 3.20 ± 0.23 | 4.64 ± 1.33 | 6.28 ± 0.90 |
| *Cyclobalanopsis gracilis* | 1× | 73.16 ± 3.34 | 12.20 ± 2.16 | 2.12 ± 1.18 | 5.08 ± 1.06 | 7.44 ± 1.02 |
| | 2× | 73.28 ± 1.79 | 10.40 ± 0.91 | 2.16 ± 1.11 | 7.40 ± 1.60 | 6.76 ± 1.32 |
| | 3× | 73.61 ± 1.09 | 11.31 ± 0.97 | 2.08 ± 0.52 | 6.52 ± 1.58 | 6.48 ± 0.28 |
| *Platycarya longipes* | 1× | 73.51 ± 2.44 | 11.69 ± 0.76 | 2.88 ± 0.47 | 4.20 ± 0.61 | 7.72 ± 2.14 |
| | 2× | 73.83 ± 4.25 | 10.37 ± 1.85 | 2.16 ± 0.72 | 6.32 ± 1.53 | 7.32 ± 0.50 |
| | 3× | 75.53 ± 3.16 | 11.11 ± 1.71 | 3.76 ± 0.58 | 4.40 ± 1.02 | 5.20 ± 0.86 |
| *T*-test | 1× | $T = 1.50, p > 0.05$ | $T = 0.84, p > 0.05$ | $T = 6.13, p < 0.005$ | $T = 8.18, p < 0.005$ | $T = 3.82, p < 0.005$ |
| | 2× | $T = 3.91, p < 0.005$ | $T = 0.25, p > 0.05$ | $T = 11.95, p < 0.005$ | $T = 6.72, p < 0.005$ | $T = 4.87, p < 0.005$ |
| | 3× | $T = 1.01, p > 0.05$ | $T = 1.52, p > 0.05$ | $T = 5.44, p < 0.005$ | $T = 9.71, p < 0.005$ | $T = 10.25, p < 0.005$ |
**Table 2** (*continued*)

| Tree species | Concentrations | The micro-aggregates at different aggregate diameters(%) | | | | |
|---|---|---|---|---|---|---|
| | | 2–0.25 mm | 0.25–0.05 mm | 0.05–0.02 mm | 0.02–0.002 mm | >0.002 mm |
| *CV* | *CV* of 1× | 2.35 | 11.54 | 37.38 | 18.28 | 8.30 |
| | *CV*$_u$ | 1.41 | 1.29 | N/A[*] | 1.28 | 1.29 |
| | *CV* of 2× | **1.26** | 11.11 | 26.75 | 22.67 | 9.84 |
| | *CV*$_u$ | **1.68** | 1.29 | 1.28 | 1.28 | 1.29 |
| | *CV* of 3× | 2.58 | 8.45 | 27.39 | 16.67 | 7.11 |
| | *CV*$_u$ | 1.39 | 1.29 | 1.28 | 1.28 | 1.30 |

**Notes.**

*N/A shows the value of *CV* great enough (*CV* > 30%) and it does not need to be tested with the *CV*$_u$.

## Active biological matter and organic matter of the root exudates of the incubated soils

The total sugar content is the highest and lowest in the rhizosphere soils of *C. pubescens* and *P. longipes,* respectively (Table 3). The highest and lowest total amino acid contents are observed in the rhizosphere soils of *C. platycarpa and L. lucidum and I. yunnanensis,* respectively. However, the highest concentrations of phenolic compounds and free amino acid are determined in the rhizosphere soils of *C. platycarpa* and *I. yunnanensis,* respectively. The contents of all active biological matter are closely related to the MWD and GMD of the rhizosphere soils of these eight plants; the total amino acid content shows the highest significance. We detected high contents of free amino acid or total amino acid in the rhizosphere soils of *C.platycarpa*, *I.yunnanensis C. gracilis,* and *P.longipes*. The rhizosphere soils of the four plants contain a high concentration of water-stable aggregates (>2 mm and 2–1 mm, Table 1) and show high MWD and GMD values (Fig. 2).

The organic matter in the root exudates identified by GC-MS primarily includes hydrocarbon, amide, alcohol, phenolic ether, aldehyde, acid, ketone, and ester (low concentrations, Table 3). Each type also includes many specific compounds (Supplemental Information 1 and Supplemental Information 2). These types of organic matter account for more than 80% of the total organic matter of the root exudates except for *C. glandulifer um* and *C. gracilis.* Hydrocarbon shows the highest percentage of all types.

The amount of specific compounds of hydrocarbon is also the highest. The amide, phenolic ether, alcohol, and ester percentages are relatively lower than that of hydrocarbon. Only four types of organic matter are closely related to MWD and GMD. Amide, aldehyde, and ester showed a higher correlation than other organic matter types. The rhizosphere soils of *C.platycarp* and *C.gracilis* contain relatively high contents of amide and ester. Correspondingly, the rhizosphere soils of the two plants comprise a high concentration for water-stable aggregates (>2 mm and 2–1 mm, Table 1) and have high MWD or GMD values (Fig. 2).

## AES and root exudates in the rhizosphere soils of additional plants

The status and degree of aggregation of the rhizosphere soils of the additional plant species are greater than that of the non-rhizosphere soils (Table 4). Conversely, the dispersion ratio and coefficient are smaller than those of the non-rhizosphere soils. The paired $t$-test indicates that the four comprehensive AES indices of the rhizosphere and non-rhizosphere soils are significantly different, except for the degree of aggregation. The status and degree of aggregation and dispersion ratio and coefficient of the additional plant species show a high variation; the *CVs* of the four AES indices are greater than the $CV_u$ values. The *CVs* of the dispersion ratio and coefficient are higher than 30% (Table 5).

Based on the CV, the variation of the active biological matter is smaller than that of the organic matter detected by CG-MS. Other statistical quantities such as the variance, standard deviation, mean, maximum, and minimum also vary. On average, the total soluble sugar and phenolic compounds are higher than the total amino acids in the rhizosphere soils of the additional plant species (Table 5). The free amino acid content is the lowest. The *CV*s of the active biological matter contents are not only greater than the $CV_u$, but also higher than 30%, indicating a significant interspecific difference. The relative hydrocarbon content

Wang et al. (2017), *PeerJ*, DOI 10.7717/peerj.3029

**Table 3 Active biological matter concentration and relative organic matter content identified by GC-MS and their correlations with MWD and GMD.**

| Tree species | Total solubl sugar (g/kg) | Total amino acids (g/kg) | Phenolic compound (g/kg) | Free amino acid (mg/kg) |
|---|---|---|---|---|
| *Carpinus pubescens* | 1.64 | 0.91 | 3.38 | 9.13 |
| *Cladrastis platycarpa* | 1.24 | 1.04 | 3.49 | 14.24 |
| *Zanthoxylum planispinum Sieb.et Zucc.* | 1.31 | 0.78 | 2.43 | 2.88 |
| *Ligustrum lucidum* | 1.11 | 0.24 | 3.39 | 12.38 |
| *Itea yunnanensis* | 1.49 | 0.24 | 2.71 | 17.97 |
| *Cinnamomum glanduliferum* | 1.51 | 0.36 | 2.76 | 12.1 |
| *Cyclobalanopsis gracilis* | 1.11 | 0.84 | 3.08 | 3.28 |
| *Platycarya longipes* 1.05 | 0.8 | 2.32 | 10.65 | |
| Pearson correlation coefficients | 0.41*; 0.48* | 0.50**; 0.55** | 0.52**; 0.48* | 0.39*; 0.44* |
| *n* | 24 | 24 | 24 | 24 |

| Tree species | Hydrocarbon (%) | Amide (%) | Alcohol (%) | Phenolic ether (%) | Aldehyde (%) | Acid (%) | Ketone (%) | Ester (%) | Other (% | Total (%) |
|---|---|---|---|---|---|---|---|---|---|---|
| *Carpinus pubescens* | 37.09 (35) | 8.61(3) | 15.36(12) | 1.45(1) | 0.72(2) | 4.35(2) | 4.93 | 6.87(3) | 5.26(7) | 84.62 |
| *Cladrastis platycarpa* | 31.85(21) | 11.91(3) | 3.10 (3) | 9.73 (3) | 1.73(4) | 0.00(0) | 1.14(1) | 27.51(4) | 1.72(4) | 88.70 |
| *Zanthoxylum planispinum Sieb.et Zucc.* | 36.72(40) | 1.25(2) | 12.24(11) | 10.63(4) | 1.07(3) | 3.70(2) | 2.11(4) | 13.75(5) | 2.94(5) | 84.40 |
| *Ligustrum lucidum* | N/A | N/A | N/A | N/A | N/A | N/A | N/A | N/A | N/A | N/A |
| *Itea yunnanensis* | N/A | N/A | N/A | N/A | N/A | N/A | N/A | N/A | N/A | N/A |
| *Cinnamomum glanduliferum* | 26.11(32) | 1.04(3) | 13.99(13) | 20.57(2) | 2.05(3) | 1.99(4) | 4.30(6) | 6.15(7) | 1.29(2) | 77.47 |
| *Cyclobalanopsis gracilis* | 31.29(50) | 7.64(4) | 9.37(11) | 2.04(4) | 1.68(3) | 0.17(1) | 8.30(8) | 5.42(5) | 3.00(9) | 68.91 |
| *Platycarya longipes* | 37.08(35) | 5.22(3) | 22.11(11) | 7.16(2) | 0.47(1) | 0.87(1) | 1.38(4) | 4.58(4) | 2.67(6) | 81.53 |
| Pearson correlation coefficients(r) | 0.39; 0.48* | 0.67**; 0.64** | −0.12; 0.08 | 0.02; 0.08 | 0.48*; 0.51* | −0.31; −0.28 | 0.27; 0.33 | 0.57*; 0.49* | 0.16; 0.20 | 0.42; 0.51* |
| *n* | 18 | 18 | 18 | 18 | 18 | 18 | 18 | 18 | 18 | 18 |

Notes.
The values in the parenthesis are the number of the specific organic matter identified by GC-MS. N/A shows no identified due to the unsatisfactory flow diagrams. *,** or *** represents a significance at a 95%, 99% or 99.9% confidence level, respectively.

**Table 4  Comparison of the AES of the rhizosphere and non-rhizosphere soils of the additional plant species.**

| Indices of the AES | Position | Average (%) | N | Standard deviation | t | P |
|---|---|---|---|---|---|---|
| Aggregation status | R | 51.17 | 42 | 14.47 | 3.014 | 0.0044 < 0.01 |
| | B | 48.04 | 42 | 14.47 | | |
| Degree of aggregation | R | 70.61 | 42 | 15.84 | 1.484 | 0.1455 > 0.05 |
| | B | 69.04 | 42 | 16.25 | | |
| Dispersion ratio | R | 36.29 | 42 | 13.25 | −3.343 | 0.0018 < 0.01 |
| | B | 40.09 | 42 | 13.23 | | |
| Dispersion coefficient | R | 25.06 | 42 | 10.04 | −2.024 | 0.04957 < 0.05 |
| | B | 28.12 | 42 | 12.95 | | |

**Notes.**
R, the rhizosphere soils; B, non-rhizosphere soils.

is the highest of all nine organic matter types identified by GC-MS. The relative alcohol and ester contents rank second and third, respectively. The $CV$ and $CV_u$ also indicate significant differences in the relative contents of these organic matter compounds of the additional 19 plant species; most of the $CV$s are significantly higher than 30%. Only hydrocarbon and the total in a relative content are slightly lower than 30%. However, both are larger than the $CV_u$ (Table 5). Comparatively, the interspecific variation of the organic matter identified by GC-MS is more significant than that of the active biological matter (Table 5).

Most of the relative organic matter contents (identified by GC-MS) of the root exudates of the rhizosphere soils of the additional plant species do not significantly correlate with the comprehensive AES indices (Table 6). Only hydrocarbon shows a significant correlation with the two comprehensive AES indices, the aggregation status and dispersion ratio. If a 90% confidence level is considered to be a weak correlation, the relative content of the phenolic ether shows significant correlation. However, almost all active biological matter correlates highly with the four comprehensive AES indices (Table 6).

The four comprehensive AES indices indicate different changes with varying contents of organic matter; the organic matter significantly correlates with AES, at least at a 90%-confidence level (Fig. 3 and Table 6). The aggregation status shows an increase with increasing hydrocarbon content, while the dispersion ratio displays a significant decrease (Fig. 3A). The values of the dispersion ratio slightly scatter as a function of the phenolic ether content; however, the dispersion ratio increases, indicating the negative effect of the phenolic ether on the AES (Fig. 3B). The comprehensive AES indices correlate relatively strongly with the active biological matter contents (Figs. 3C–3E). Specifically, both the status and degree of aggregation present logarithmic or linear growth with increasing active biological matter content. Consequently, the dispersion ratio and coefficient show a logarithmic decrease or change of the power function. These organic matter compounds explain 20%–76% of the variation in the total effect of the root exudates on the AES based on the different $R^2$ values. The phenolic compound can explain the aggregation status the best. It is worth noting that although Fig. 3F was used to describe the relationship between the free amino acid contents and comprehensive AES indices, it also represents the varying characteristics of other insignificant comprehensive AES indices corresponding to the organic matter contents (Table 6).

**Table 5  Statistical quantities of the comprehensive AES indices, active biological matter contents, and relative concentrations of the organic matter identified by GC-MS in the rhizosphere soils of the additional plants.**

| Indices of the AES | Aggregation status | Degree of aggregation | Dispersion ratio | Dispersion coefficient |
|---|---|---|---|---|
| Maximum (%) | 71.09 | 85.77 | 71.10 | 44.69 |
| Minimum (%) | 21.51 | 37.35 | 16.93 | 10.45 |
| Mean ($\mu$, %) | 51.17 | 70.61 | 36.29 | 25.06 |
| Variance ($\sigma$) | 209.49 | 250.89 | 175.69 | 100.80 |
| Standard deviation ($s$) | 14.47 | 15.84 | 13.25 | 10.04 |
| Coefficients of variation ($CV$) | 28.29 | 22.43 | 36.52 | 40.06 |
| $CV_u$ | 1.17 | 1.17 | 1.16 | 1.16 |
| $n$ | 42 | 42 | 42 | 42 |

| Biological active matters | Total soluble sugar | Total amino acids | Phenolic compound | Free amino acid |
|---|---|---|---|---|
| Maximum (g/kg) | 2.26 | 1.04 | 3.81 | 0.01835 |
| Minimum (g/kg) | 0.27 | 0.17 | 0.81 | 0.00286 |
| Mean ($\mu$, g/kg) | 1.06 | 0.56 | 2.52 | 0.01010 |
| Variance ($\sigma$) | 0.23 | 0.09 | 0.82 | 0.00002 |
| Standard deviation ($s$) | 0.48 | 0.29 | 0.90 | 0.00397 |
| Coefficients of variation ($CV$) | 45.75 | 52.32 | 35.83 | 39.33 |
| $CV_u$ | 1.16 | 1.16 | 1.16 | 1.16 |
| $n$ | 42 | 42 | 42 | 42 |

| Parameters | Hydrocarbon | Amides | Alcohols | Phenolic ether | Aldehyde | Acids | Ketone | Esters | Others | Total |
|---|---|---|---|---|---|---|---|---|---|---|
| Maximum (%) | 49.66 | 11.91 | 33.55 | 20.57 | 2.61 | 21.47 | 11.70 | 27.51 | 5.26 | 92.01 |
| Minimum (%) | 18.98 | 0.44 | 3.10 | 1.12 | 0.23 | 0.16 | 1.04 | 4.12 | 0.55 | 67.28 |
| Mean ($\mu$, %) | 37.27 | 4.36 | 14.58 | 6.35 | 1.25 | 3.16 | 4.23 | 8.69 | 2.78 | 82.11 |
| Variance ($\sigma$) | 65.43 | 10.54 | 47.57 | 40.71 | 0.41 | 23.83 | 7.62 | 31.22 | 1.95 | 50.45 |
| Standard deviation ($s$) | 8.09 | 3.25 | 6.90 | 6.38 | 0.64 | 4.88 | 2.76 | 5.59 | 1.40 | 7.10 |
| Coefficients of variation ($CV$) | 21.70 | 74.41 | 47.30 | 100.47 | 50.92 | 154.70 | 65.31 | 64.27 | 50.28 | 8.65 |
| $CV_u$ | 1.20 | 1.20 | 1.20 | 1.20 | 1.20 | 1.20 | 1.20 | 1.20 | 1.20 | 1.21 |
| $n$ | 19 | 18 | 19 | 19 | 19 | 17 | 19 | 19 | 19 | 19 |

**Notes.**

$n$, the number of plant species.

Peer J

**Table 6  Pearson coefficients for the correlations between the four comprehensive AES indices and the relative contents of organic matter identified by GC-MS and the active biological matter in the rhizosphere soils of the additional plant species.**

| Types of organic matter | | Aggregation status | | | Degree of aggregation | | | Dispersion ratio | | | Dispersion coefficient | | |
|---|---|---|---|---|---|---|---|---|---|---|---|---|---|
| | | r | p | n | r | p | n | r | p | n | r | p | n |
| Organic matter identified by GC-MS | Hydrocarbon | **0.47** | <0.05 | 19 | 0.31 | >0.10 | 19 | **−0.46** | <0.05 | 19 | −0.03 | >0.10 | 19 |
| | Amides | −0.30 | >0.10 | 19 | −0.23 | >0.10 | 19 | 0.29 | >0.10 | 19 | −0.2 | >0.10 | 19 |
| | Alcohols | −0.04 | >0.05 | 19 | −0.27 | >0.10 | 19 | −0.06 | >0.10 | 19 | −0.22 | >0.10 | 19 |
| | Phenolic ether | −0.05 | >0.05 | 19 | 0.15 | >0.10 | 19 | 0.12 | >0.10 | 19 | **0.39** | **<0.10** | **19** |
| | Aldehyde | −0.13 | >0.05 | 19 | 0.05 | >0.10 | 19 | 0.16 | >0.10 | 19 | −0.11 | >0.10 | 19 |
| | Acids | 0.22 | >0.05 | 19 | 0.08 | >0.10 | 19 | −0.23 | >0.10 | 19 | 0.01 | >0.10 | 19 |
| | Ketone | −0.18 | >0.05 | 19 | −0.21 | >0.10 | 19 | 0.13 | >0.10 | 19 | −0.06 | >0.10 | 19 |
| | Esters | 0.22 | >0.05 | 19 | 0.16 | >0.10 | 19 | −0.19 | >0.10 | 19 | 0.01 | >0.10 | 19 |
| | Others | 0.25 | >0.05 | 19 | 0.24 | >0.10 | 19 | −0.23 | >0.10 | 19 | −0.2 | >0.10 | 19 |
| Active biological matter | Total sugar | **0.75** | <0.001 | 42 | **0.71** | <0.001 | 42 | **−0.63** | <0.001 | 42 | **−0.27** | **<0.10** | **42** |
| | Total amino acids | **0.62** | <0.001 | 42 | **0.57** | <0.001 | 42 | **−0.57** | <0.001 | 42 | **−0.28** | **<0.10** | **42** |
| | Phenolic compound | **0.80** | <0.001 | 42 | **0.87** | <0.001 | 42 | **−0.58** | <0.001 | 42 | −0.32 | <0.05 | 42 |
| | Free amino acid | 0.13 | >0.10 | 42 | 0.14 | >0.10 | 19 | −0.07 | >0.10 | 19 | −0.18 | >0.10 | 19 |

**Notes.**

The values in bold type represent significant relationships at a 90%, 95%, 99% or 99.9% confidence level respectively, when $p < 0.10$, 0.05, 0.01, or 0.001.

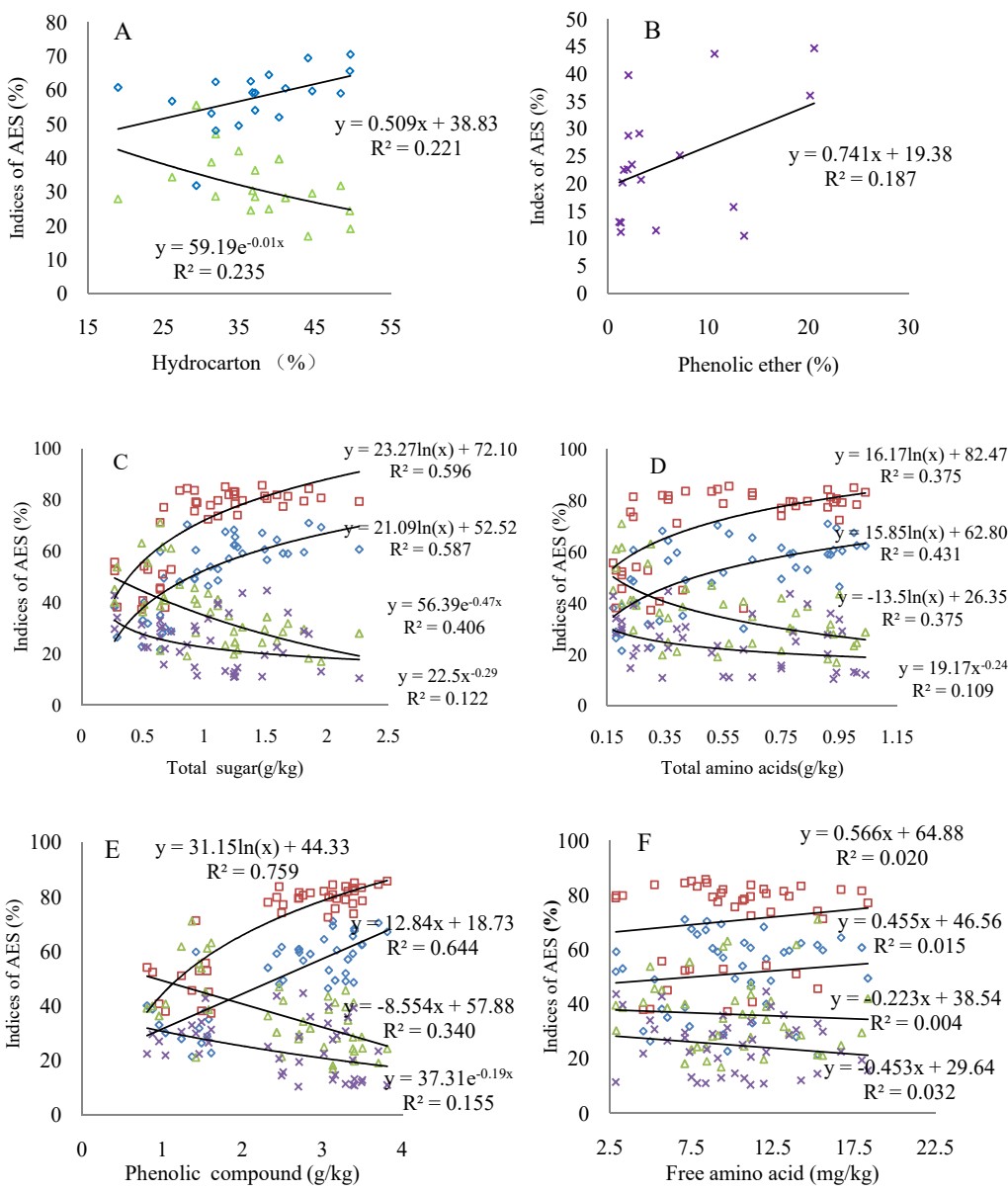

**Figure 3  Changes of the comprehensive AES indices with varying key organic matter contents.** (A) Hydrocarbon; (B) phenolic ether; (C) total sugar; (D) total amino acids; (E) phenolic compound; (F) free amino acid; triangular symbols: dispersion ratio; square symbols: degree of aggregation; diamonds: aggregation status; multiplication sign: dispersion coefficient. The regression models with $R^2 > 0.20$ indicate the confidence level of 95% ($p < 0.05$) based on the goodness of fit test. The number of observations in all figures is the same as that in Table 6.

# DISCUSSION

The direct effect of the root exudates on the AES is often tested using an incubation of mixtures of soil samples and root exudates from few annual plants (*Song et al., 2009a*; *Song et al., 2009b*). *In situ* experiments are also conducted based on the exclusion of the effect of the root systems, plant cover and litters to quantify the effect of the root exudates

(*Wang et al., 2014a*; *Wang et al., 2014b*). We collected the root exudates of eight woody plant species and performed an incubation experiment to identify the responses of the water-stable aggregates, microaggregates, MWD and MGD of the soil samples to different root exudates. The results indicate that the water-stable aggregates (>2 mm and 2–1 mm), MWD, and MGD increase (Table 1). Conversely, the microaggregates (0.05–0.02 mm and <0.002 mm, Table 2) and water-stable aggregates (<0.25 mm, Table 1) notably decline. The increases of the water-stable aggregates (>2 mm and 2–1 mm) ranges on average from 15.52% to 40.65% and that of MWD and MGD from 8.58%–16.41% compared with the control samples. However, the increase is smaller than that observed in previous studies in which the root exudates were simulated using analogues of the root exudates (*Traoré et al., 2000*) and were collected from the rhizosphere soils of soybean and maize (*Song et al., 2009a*; *Song et al., 2009b*). Because the soil incubated in this study is a rendzina soil with coarse silt (0.02–0.05 mm), the smaller soil particles only account for17.34% of the soil, far less than 84.2% and 59% in the incubated soils collected from luvisol (*Traoré et al., 2000*) and black soil (*Song et al., 2009a*; *Song et al., 2009b*), respectively. This was notably unfavorable with respect to adhering soil particles to form water-stable aggregates.

The AES of the rhizospheres of the additional plant species in the extrapolation experiments, quantified based on the status and degree of aggregation and the dispersion ratio and coefficient, is different from that of the non-rhizospheres (Table 4 and Supplemental Information 5). These four indices of the rhizosphere soils also indicate a strong difference among these plant species (Table 5 and Supplemental Information 6). The interspecific differences are greater than those in the incubation experiments based on the $CV$ and $CV_u$ (Tables 1, 2 and 5). Previous studies have not paid attention to the interspecific differences with respect to the woody plants that enhance the AES with their root exudates; the interspecific differences were determined in this study using $CV$ (*McCully, 1999*; *Czarnes et al., 2000*; *Whalley et al., 2005*; *Young & Crawford, 2004*; *Wang et al., 2014a*; *Wang et al., 2014b*). Previous studies also suggested that the impact of the root exudates on the physical properties and structure of the soil still has to be deciphered; thus, a series of structured repacked samples was incubated with a daily input of artificial root exudates (*Milleret et al., 2009*; *Kohler-Milleret et al., 2013*). The results indicate that the root exudates increase the microbial activity and aggregate stability and decrease the small-diameter structural porosity. This study enhances the understanding of the impact of the root exudates on the physical soil properties based on the AES responses to the root exudates described by the different indices and interspecific differences. The results of this study were obtained for a karst region. The loss of soil particles due to seepage flow, which means concealed erosion, is a primary factor in karst regions. Ecologists can apply the plant species with rhizosphere soils that have greater aggregation status and degree values to ecological engineering to reduce the concealed erosion due to seepage flow.

Root exudates are defined as diffusible compounds in which free sugars, amino acids, and organic acids have been widely recognized to not only have adhesive effects (*Jones, Nguyen & Finlay, 2009*) but also a variety of active biological effects (*Whipps, 2001*; *Song et al., 2009a*; *Song et al., 2009b*). Soil microbes require active biological matter; the growths of the soil microbes can result in rich myceliums promoting the formation of water-stable

aggregates (*Whipps, 2001*). In this study, we tested the concentrations of four types of active biological matter of the root exudates from eight plant species. We further used GC-MS to identify the specific organic compounds and analyzed the correlations of the organic matter with the AES (*Morel et al., 1987*; *Gessa & Deiana, 1990*; *Albalasmeh & Ghezzehei, 2014*). The organic matter includes hydrocarbon, amides, alcohols, phenolic, aldehyde, acids, ketone, esters and other low-concentration matter. It accounts for~80% of the organic matter of the root exudates; the relative hydrocarbon content is the highest. The active biological matter, hydrocarbon, amides, aldehyde, and ester detected by GC-MS are primarily responsible for the changes of the MWD and GWD (Table 3).

To further validate the effect of the root exudates in the incubation experiments, we measured the organic matter content of the root exudates extracted from the rhizosphere soils of the additional plant species. The relative amount of organic matter and interspecific variation is beyond our expectations (Supplemental Information 1 and Supplemental Information 2). The active biological matter contents show less interspecific variation than the organic matter identified by GC-MS (Table 5 and Supplemental Information 6). The comprehensive AES indices, aggregation status and degree and dispersion ratio and coefficient of the rhizosphere soils of the additional plant species are primarily related to the hydrocarbon, amide, phenolic ether, total sugar, total amino acid and phenolic compounds (Table 6 and Supplemental Information 7). These organic compounds are crucial to the AES, which is similar to the results of the incubation experiments. Most of the organic matter detected by GC-MS is not significantly associated with the comprehensive AES indices. These organic compounds may be the hormone-like compounds of low molecular fractions affecting the growth and nutrient uptake of other plants and microbes in addition to allelopathic indirect effects on the AES (*Nardi et al., 2005*). The regression analysis indicated that the key organic matter determined by correlation analysis can explain 20%–76% of the variation in the total effect of the root exudates on the AES (Fig. 3).

## CONCLUSIONS

The water-stable aggregates, MWD and GMD of the soils incubated with root exudates significantly increase. The concentration of most of the microaggregates and small water-stable aggregates decreases. The root exudates of *C. platycarpa*, *C. gracilis*, *I. yunnanensis,* and *P. longipes* cause a relatively higher increase of the amount of water-stable aggregates (>2 mm and 2–1 mm), MWD and GMD than that of other plants. The root exudates contain hundreds of organic matter compounds. Total sugar, total amino acids, phenolic compound, hydrocarbon, amides, and phenolic ether are crucial to soil aggregation and AES; however, phenolic ether has a negative effect on the AES. The organic matter detected by GC-MS shows a great interspecific difference compared with the active biological matter; however, the active biological matter has a higher correlation with the AES. The changes of the comprehensive AES indices with varying key organic matter content differ. The interspecific differences of the AES of the rhizospheres affected by the key organic matter of the root exudates provide an opportunity to reduce the loss of soil particles due to the seepage flow and enhance the soil stability in karst regions.

### Funding

This work was supported by grants from the National Natural Science Foundation of China (No. 40861015), the Foundation of State Key Laboratory of Soil Erosion and Dryland Forming on Loess Plateau (No. 10501-161). The funders had no role in study design, data collection and analysis, decision to publish, or preparation of the manuscript.

### Grant Disclosures

The following grant information was disclosed by the authors:
National Natural Science Foundation of China: No. 40861015.
Foundation of State Key Laboratory of Soil Erosion and Dryland Forming on Loess Plateau: No. 10501-161.

### Competing Interests

The authors declare there are no competing interests. Hong Fang is an employee of Water-affair Authority of Xifeng County, Guiyang, Guizhou, China.

### Author Contributions

- Zhen Hong Wang conceived and designed the experiments, contributed reagents/materials/analysis tools, wrote the paper, prepared figures and/or tables, reviewed drafts of the paper.
- Hong Fang performed the experiments, analyzed the data, contributed reagents/materials/analysis tools, wrote the paper, prepared figures and/or tables.
- Mouhui Chen performed the experiments.

### Field Study Permissions

The following information was supplied relating to field study approvals (i.e., approving body and any reference numbers):

All field experiments in the study were approved by Administration Bureau of Two Lakes and One Reservoir in Guiyang City (ABTLORGC).

### Data Availability

The raw data has been supplied as a Supplementary File.

### Supplemental Information

Supplemental information for this article can be found online at http://dx.doi.org/10.7717/peerj.3029#supplemental-information.

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
