# Peer review of "Effects of root exudates of woody species on the soil anti-erodibility in the rhizosphere in a karst region, China"

_PeerJ, doi:10.7717/peerj.3029_

## Round 0.1 · original submission · Major Revisions

Both reviewers found that this work has merit in publication. However the authors are required to revise the manuscript thoroughly and clarify the methods. Proof-reading by an English speaker would be beneficial.

Reviewer 1 ·

Basic reporting

The research presented in the manuscript concerns the effect of root exudates on the anti-erodiblity of soils in the rhizosphere. It was interesting that the authors examined this effect in greater detail by studying different types of organic matter. The structure of the manuscript is well-ordered except that there is no separate conclusion section. Between the different parts of the manuscript there is some repetition. E.g. line 115-129 should be part of the material and methods and should not be included in the introduction.

In general as a reader, the manuscript was difficult to read as there are some issues with the used English language. Therefore I would suggest to get some expert help in editing the English language before resubmitting the manuscript. Some major issues are: 1) the use of too long sentences of 5 lines with a lot of clauses. As a result one has to read some sentences 3 or 4 times to understand the message: e.g. line 85-89; 96-101; 115-118; 118-122; 124-129; 222-228; 312-316; 434-437, 2) the manuscript is too wordy with a lot of unnecessary information and repetitions and 3 ) lot of errors concerning the prepositions and word order.

Literature is well referenced however several typos occurred: Line 71: Fattet et al. 2011; Vannoppen et al., 2015 (see also line 68, 81), line 486 De Baets et al., 2008. See also references line 527 De Baets S; line 536 Fattet; Line 594 Vannoppen W, Vanmaercke M, De Baets.

The introduction and background provide a lot of information of previous research and the objectives are clear. However, I would highlight more that the research was done in a karstic environment as this is now only clear after reading the discussion section. This should also be highlighted in the title. A suggestion could be: Effects of root exudates of woody species on the soil anti-erodibility in the rhizosphere in a karst region, China.

A lot of information is provided in the tables which is not explicitly discussed in the manuscript. Maybe it is a possibility to include some tables or parts as supplementary information.

Figure 1: explain the used abbreviations in the graph in the caption of the figure.

Figure 2 is difficult to read as there is too much information provided in the graphs that it is not readable anymore. The y-axis is not labeled. A suggestion could be indices of AES (%). For every single graph I would include a legend for the used symbols. E.g. 2A what are the triangles indicating? All equations are put in the graph. To increase the readability I would suggest to place them in a separate table, together with R², number of observations and the p-value of the relationship. X-lable graph 2A: Hydrocarbon, x-lable graph 2C: total soluble sugar. Caption of figure two should be improved: B:Amides should be removed.

Experimental design

The problem statement is well structured and the research gaps are clear. However, no clear research questions were defined, which should be included in the improved manuscript.

The used methods are very clear described with sufficient information to be reproducible by another investigator. However, this section is too wordy by using a lot of repetitions and redundant information which makes it more difficult to read.
For the data analysis both t-tests as well as the coefficient of variation were used. Somehow, the data analysis is somewhat limited. The authors mention in the discussion that the information can be used to select tree species for enhancing AES, isn’t it possible to test which species of the selected ones in the study are the most effective in enhancing AES? This would provide relevant information for practitioners in the field.

Three different analyses were carried out: 1) effect root exudates on AES for 8 species, 2) GC-MC analysis and test of biological active matter for 8 species and 3) for 34 species. I was wondering if the authors compared the results of part 1 (table 1, fig 1) and the findings of part 2 (table 3 and 4). Or in other words, can the observed differences in MWD and GMD be explained by the observed differences in contents of biological active matters? Did the authors also measured the indices of the AES for the eight species? If so, these data could be used to validate the findings in table 9.

Table 9 / Figure 2: Which correlation test was used to calculate the r-values? These r-values give the amount of correlation for a linear relationship however most relationships are non-linear. So transformations of the variables can lead to higher r-values and maybe more significant coefficients of correlations. In graph 2B and 2C a quadratic polynomial is suggested as relationship. This means that for 2B the dispersion coefficient will increase with decreasing phenolic ether below 5%. The same remark for graph 2C in which the dispersion ratio will increase with increasing total soluble sugar.

Some general comments on the material and methods section are:
Line 146-148, this is unclear formulated. Did the authors take 1 or more subsamples of 60g before or after drying the initial 500g of soil? Line 156-157 for readers who are not familiar with the soil type rendzina it is difficult to understand. I suggest to rewrite this sentence: Samples were taken from a rendzina soil and incubated … . Line 180-185: mention the units. Line 235-236 typo: ‘tested value’. Line 275-277: how can this be observed in the table?

Validity of the findings

The data presented in the manuscript are robust, statistically sound and controlled. No replication experiments were done for the 34 species which should be increased the findings in this manuscript. The conclusion is now included in the discussion section. Therefore, I would suggest to add a conclusions section in which the answers are given on the states research questions.

Additional comments

No Comments

Reviewer 2 ·

Basic reporting

The authors investigate the influence of organic compounds in the rhizosphere of woody tree species on the aggregate size distribution and aggregate stability (inverse of aggregate dispersion) of a rendzina soil. Aim of the study is to investigate the anti-erodibility of soils (AES) as influenced by tree root exudates.
They conduct an extensive experiment, where soil samples from the rhizospheres of eight tree species were taken in a karst forest in the Southwest of China. From these soil samples root exudates were extracted for each of the species. A rendzina soil was incubated with root exudates from the single tree species, and aggregation status of the soil was measured before and after incubation.
Additionally, soil of the rhizospheres of another 34 tree species was sampled, and the aggregation status of these soils was measured.
Biological active compounds were measured for all rhizosphere soil samples using chemical analytical methods Organic matter compounds were analyzed for six of the eight rhizospheres and for 13 of the 34 additional sampled rhizospheres, using GC-MS.
While in general it is interesting research worth being published there are some issues the authors should work on to improve the readability of the manuscript.
(1) There are nine tables in the manuscript, and the text of the result section mainly describes the findings shown in theses tables. Some information is presented double, for example results from the eight tree species are later included into the so-called “interpolated” tree species. It would be helpful for the reader to condense the number of tables to not more than six, and instead of describing the results from the tables in the text focusing more on interpreting the results towards the effect of specific root exudates from specific trees on the AES as quantified by aggregate size distribution and stability. As a suggestion, Tables 1 and 2 could be merged, and Tables 3 and 4, and Tables 7 and 8.
(2) The whole text is a bit lengthy and thus difficult to follow. I suggest shortening it to a total length of about 2/3.
(3) In general, the English is good, but there are flaws in the text, and some of them are continuously repeated. One of it is the use of “organic matters” which should be called “organic matter” or “organic matter compounds”, with matter in the singular. For gas chromatography – mass spectrometer use the abbreviation GC-MS throughout, not GC-MC.

Experimental design

A figure with a flowchart describing the experimental design would be very helpful. I had to go through the text back and forth to understand what measurements the authors had performed.
The term “extrapolated plant species” is misleading and should not be used here. The 34 additional tree species could for example be called “additional plant species”.
In the tables and in the text it is not always clear where the number of plant species came from: eight species were investigated initially, plus 34 additional plant species, yielding a total of 42. For 13 from the 34 additional plant species results on the organic matter compounds of the rhizosphere could be obtained by using the GC-MS (line 224-225). Then the number 19 in tables 8 and 9 comes from six out of nine plus 13 out of 34 GC-MS data sets available?

Validity of the findings

To the best of my knowledge, the findings are valid. As mentioned before, the authors should focus more on interpreting these findings, for example discussing the relationship between specific biological active substances and organic matter compounds with high values (Table 3 and 4), and how these relate to aggregate diameter observed for the rendzina incubated with root exudates from specific plants (Figure 1).
It would be good to provide the results from Table 4 and 8 also in g/kg.
I am not sure whether the raw data provided for the GC-MS analysis (supplemental Tables 1 and 2) are sufficient to meet the policy of the journal.

Additional comments

A scanned version of the annotated manuscript is attached, instead off listing my comments to single lines of the text. In general, encircled line number mean a change is suggested. Wavy underlined text refer to a change in style (wording), as well as an encircled capital letter A close to the line number. Text that is underlined with a straight line I considered important, in most cases no change is needed there. A question mark means it is not clear what the authors want to say.
A few important comments line by line:
Line 133: please provide the altitude of the experimental karst forest, and the name of the Chinese province.
Line 165: what volume (in ml) of the mother liquid was applied?
Lines 179 and 182: I am not familiar with using this kind of geometry mean. Usually, the geometric mean is estimated from a product and taking the nth root of it (see for example the web site of MathWorks).
When the authors use an exponent and logarithm instead, they should provide a citation where this equation comes from.
Line 238: what soil dispersant has been used, and in which volume and molarity?
Line 252-255: please provide a citation for estimation of the CVU.
Figure 3: the figure caption does not match the figure. For A, the legend for the green triangles is missing. Figure 2B, percentage of AES as a function of amides (mg/kg) is missing, but it is written in the caption. This shifts all the captions one letter too far.
Line 422: here it should be discussed in more detail for which exudates (from which plant species) the percentage of water-stable aggregates increased.
Line 510: see my comment above: which plant species are able to increase the AES?
The supplemental Table 2 is not cited in the text.

Annotated reviews are not available for download in order to protect the identity of reviewers who chose to remain anonymous.

---

## Round 0.2 · Minor Revisions

The paper has been revised according to the reviewers' comments. I concur with both reviewers that the English should be further improved to ensure that your international audience can clearly understand your
text. I suggest that you seek help from a native English speaking colleague. Some examples where the language could be improved include abstract, line 56 (what is the stability of rhizosphere?), line 57-58 (trait is not an appropriate word), etc. The current phrasing in many sentences makes comprehension difficult.

Please also address all other reviewer comments.

Reviewer 1 ·

Basic reporting

The research presented in the manuscript concerns the effect of root exudates on the anti-erodibility of soils in the rhizosphere. It was a pleasure to notice that all major issues of the previous version of the manuscript were solved. The manuscript is well written, a final conclusion section is added and all figures and tables are easily to read.

The introduction and background provide a lot of information on previous research and the objectives are clear.

The resubmitted manuscript was well written and attention was paid to the given remarks. However, few mistakes are still present which should be solved:

Line 47: … is a zone … which is affected by the zone? This is unclear formulated.
Line 70: remove ‘a’ in reference
Line 74: add a space after xanthan,
Line 77: add a space in the reference
Line 79: do the authors mean ‘were involved’ instead of ‘was implicated’?
Line 120: remove ‘the’ before incubation experiments
Line 104: research question 1 should be reformulated: eg. …, how does this affect the water-stable aggregates, … of these soil subsamples?
Line 110-111: research question should be reformulated: eg. How does the key organic matter affect these comprehensive indices of the AES?
Line 134-135: do the authors mean at least instead of no less than?
Line 146: Then, we respectively weighted three soil samples of 30 g, … .
Line 152: add a space between C and in.
Line 188: We selected 34 additional tree species … .
Line 254: Figure 2b: 3x misses in the legend of the graph.
Line (below table 6): The indices … . This is unclear formulated.
Line 375: remove ‘the’ before biological active effects.
Line 403: replace and by , before I. yunnanensis
Line 403: add a space between longipes and resulted

Experimental design

The problem statement is well structured and the research gaps are clear. Clear research questions were defined. The used methods are very clear described with sufficient information to be reproducible by others.

A suggestion is to add the type of correlation coefficient (=pearson) to the caption of table 3 and 6 to avoid any confusions. Eg. Table 6: Pearson correlation coefficients between … .

Validity of the findings

The data presented in the manuscript are robust, statistically sound and controlled.

At line 99-101, the authors mentioned the importance of the relationship between AES and root exudates for controls of concealed erosion. However, this was not further mentioned in the discussion section or in the conclusions. It would be nice to add some more information on the effects on concealed erosion in the final version of the manuscript.

Reviewer 2 ·

Basic reporting

The use of the English language is better now, but some parts of the text are still difficult to follow and understand. One reason is too long sentences which should be broken into two; another reason is the incorrect use of words or expressions as the subject or object (in a grammatical sense) in some sentences. Some examples are provided below, but it is not possible for me as the reviewer to correct the grammar in the whole document. Please make another effort to improve the writing. It would be a pity if the very interesting results from these study are somewhat hidden in a difficult to read and follow text.

The authors followed the suggestions of my first review and improved the manuscript accordingly. One of these suggestions was to include a flowchart showing the experimental design. This flowchart could be structured even better, with providing one box for every method used. Methods applied should be mentioned explicitly, for example "GC-MS". The term “extrapolated” is still used and the number of the additional tree species should be written. Instead of “species” write “tree species”. In the best case the flowchart will provide an overview for the whole experimental setup, thus catching the reader’s eye and drawing attention to read the whole document in detail.

Experimental design

no comment

Validity of the findings

no comment

Additional comments

Comments in detail, line numbers refer to final view of the “track changes” WORD document.
Line 26-27: if I understand it right you want to say “However, scientists still remain unclear regarding (1) the key organic matter in the root exudates to affect the AES and (2) interspecific variation of these root exudates.”
Line 41: write “Different plants secrete different relative contents”… without the article “the”.
Line 55-59: this sentence contains a lot of information and should be broken into two sentences.
Line 63-64: perhaps you want to say “Root exudates have a significant effect on the anti-erodibility of soils (AES), and they are influenced by factors such as root density, litter and vegetation cover.”
Line 65-67: there are too many definite articles “the” in this sentence. Write “Root exudates play several roles in directly and indirectly strengthening the AES: (1) The adhesive properties of root exudates bind soil particles together to enhance the formation of water-stable aggregates”…
Line 67: “(2) the release of root exudates”…
Line 68-124: check carefully for the use of the article “the”. It is used much too often.
Line 101: rewrite this sentence, for example start with “Understanding the relationship between AES and root exudates has a special significance for the control of concealed erosion which is a phenomenon of vertical movement of soil particles with a seepage flow but little surface runoff. It occurs in karst regions.”
Line 156: how was the respective C mass determined?
Line 228-303: are these really particle size diameters, or rather aggregate size diameters?
Line 244-299: why are there so many empty lines in the manuscript?
Line 304-313: this paragraph is difficult to read
Lines 305, 309-310: write “The T-test indicated significant differences” instead of “T-test indicated significantly different”.
Lines 391, 403: here Fig. 2 should be cited instead of Fig. 1.
Line 397: write “The specific matter of the hydrocarbon was also highest.”
Line 414-415: write “Conversely, dispersion ratio and dispersion coefficient were smaller than those of the non-rhizosphere soils.”
Line 434: write (if this is what you want to say) “On average, the total soluble sugar and phenolic compounds were higher than the total amino acids”…
Line 446: write “significantly”.
No line numbers available here, below Table 6: “The values of the dispersion ratio appeared a bit scattered”…
No line numbers available here, below Table 6: “Consequently, the dispersion ratio and coefficient presented a logarithmic”…
I do not understand the meaning of the last sentence on page 14, starting with “It was noted that”…
Line 488: write “We further used GC-MS to identify”…
Line 493: there is a typo. Write “amides” instead of “mide”.
Line 514: write “MWD” instead of “WMD”.
Line 517: write “compared to” instead of “compared with”, here and throughout the text.
Line 518-520: I do not understand the last two sentences, and it would be good if you end the article with a strong statement. Please re-write.

---

## Round 0.3 · accepted · Accept

The manuscript has been fully revised. It is now accepted for publication